# Attacks on Rollups

Adrian Koegl
adrian@quantstamp.com
Quantstamp, Inc.
USA

Zeeshan Meghji
zeeshan@quantstamp.com
Quantstamp, Inc.
USA

Donato Pellegrino
donato@quantstamp.com
Quantstamp, Inc.
USA

Jan Gorzny
jan@quantstamp.com
Quantstamp, Inc.
USA

Martin Derka
martin@quantstamp.com
Quantstamp, Inc.
USA

## ABSTRACT

A rollup is a network, implemented via smart contracts on a blockchain, that aims to scale that slow but general purpose blockchain. The rollup executes transactions and posts the resulting state root, along with the transaction data, to a blockchain they are built on. As a result, the state root of the rollup network is always recorded on the underlying blockchain. The underlying blockchain is used to derive the state of the rollup itself, meaning that the rollup state cannot be changed arbitrarily or would be easily detected (subject to how its state is updated and recorded on the underlying blockchain). In turn, the rollup inherits some security from its underlying blockchain — but the rollup network itself is not immune to direct attacks. Some attacks are like other network-level attacks (e.g., denial-of-service attacks) while others are a result of the rollup's connection to its underlying blockchain (e.g., re-organization attacks). In this work, we collect a list of known attacks on rollups and illustrate their impact.

## CCS CONCEPTS

• **Computer systems organization** → **Dependable and fault-tolerant systems and networks**; • **Security and privacy** → *Network security*.

## KEYWORDS

Rollup, blockchain, Ethereum, scaling solution, security, layer two

**ACM Reference Format:**
Adrian Koegl, Zeeshan Meghji, Donato Pellegrino, Jan Gorzny, and Martin Derka. 2023. Attacks on Rollups. In *4th International Workshop on Distributed Infrastructure for the Common Good (DICG '23), December 11–15, 2023, Bologna, Italy*. ACM, New York, NY, USA, 6 pages. https://doi.org/10.1145/3631310.3633493

## 1 INTRODUCTION

Modern blockchains like Ethereum [1] are general purpose distributed ledgers that have evolved beyond merely sending digital

*DICG '23, December 11–15, 2023, Bologna, Italy*

ACM ISBN 979-8-4007-0458-1/23/12.
https://doi.org/10.1145/3631310.3633493

assets between accounts. They often support so-called *smart contract* [2], which are programs deployed on the network and execute their code when interacted with via a transaction. Sometimes, these are simple programs like digital tokens (e.g., ERC-20 tokens [3]), and other times they more complicated, like trading applications or games.

However, these networks are not necessarily performant: Ethereum can only handle tens of transactions per second [4]. As a result of this performance bottleneck, some of the applications built on top of these blockchains aim to scale the blockchain itself [5]. For example, one can implement protocols like state channels [6–9], which allow participants to transact off of the blockchain and settle their balances only when the other party is not cooperating or when the channel should be closed. However, this particular approach is not particularly popular because a specific channel is necessary for every pair of participants, though it forms the basis of the Bitcoin Lightning network.

Other scaling protocols have also been designed and implemented. Side-chains [10] relax need for a separate setup for every pair of participants and add support for off-chain smart contract execution, but are not necessarily tied to the main blockchain; as a result, the side-chain can effectively do what it wants. So-called Plasma chains [11–13] were one approach that tried to bring the benefits of a side-chain while maintaining some dependency on the underlying blockchain itself. However, these were plagued by issues related to data availability (e.g., malicious operators could withhold data necessary to prove a withdrawal is valid).

To overcome these issues, data was required to be posted on-chain and this ultimately lead to the concept of a *rollup* (also called a *commit-chain* [14] or *validating bridge* [15]; see also [16]). A rollup is a smart-contract based protocol where transactions are executed off-chain to form another blockchain network. In a rollup, batches of the executed transactions, along with the resulting state root of network, are posted on the underlying layer one. Rollups separate execution of state transitions (that is, processing of transactions) from consensus, and in turn, they can process many more transactions per second than their underlying layer one. Since rollups post a summary of the rollup state (the root of a Merkle tree [17]) onto the underlying blockchain, which is actually also used to derive the state of the rollup network, they inherit some security from the underlying blockchain. In order to derive a different layer two chain, it is necessary to change the recorded layer two state on layer one, a challenging and expensive task. Rollups, and other scaling solutions built via smart contracts, are called "layer two"

(or "L2") solutions, built on top of a corresponding "layer one" (or "L1") blockchain.

Rollups also support smart contracts. As a result, applications can be built on them and they can be seen as middleware between such applications and the underlying layer one. Rollups can add features to, or remove features from, their virtual machine when compared to the virtual machine of the underlying layer one itself. They may also support an entirely different virtual machine, as in the case of the StarkNet rollup [18].

In addition to these possible differences on rollups, they include *bridging* functionality from the layer one to the layer two network implemented by the rollup. Digital assets are locked in a layer one smart contract and a message is relayed to the rollup network to mint a representation of the asset on the layer two. In the last two years, over $2 Billion USD was stolen from or locked in bridge contracts through bugs and attacks [19]. As rollups implement more complicated versions of bridges, it is important to understand what attack vectors are possible for these systems before they are built.

This work investigates the security consideration of these networks. These networks do not have the same attack vectors as other (layer one) blockchain networks. First, most users do not (and cannot) run nodes for these networks. They are often centralized, earning user trust through two popular approaches (defined in detail in Section 2). *Optimistic* rollups post state roots which can be challenged if they are incorrect up within a certain time period. *Zero-Knowledge (ZK)* rollups provide a mathematical proof that the update is correct. Second, while some security is inherited from the underlying layer one, its presence also complicates the layer two network itself, e.g., by re-organizing itself. Finally, the design of these systems themselves may introduce new avenues for attacks that are not relevant to layer one blockchains.

Rollup users expect liveness from the system, that is, as long as the underlying layer one is operating, the layer two is operating as well. They also expect that any bridged digital assets are (nearly) as secure on the rollup network as on the layer one itself. Some unique features of rollups, like so-called *escape hatches*, aim to make those assumptions a reality [20]. However, rollups are fairly new developments and are not always feature complete. Therefore it is also important to study the attacks on these systems in the context of their required features, to ensure that users can check if such functionality exists and determine the riskiness of the system for themselves. This work fills a gap in the current literature by providing a list of attack vectors unique to rollups. We also suggest areas of future research in order to mitigate some of these attacks and other risks on rollups.

## 2 PRELIMINARIES

A rollup can be broken down into several components[1]: a *sequencer*, a *state proposer*, and an (explicit or implicit) *verifier*. A sequencer is responsible for ordering layer two transactions and committing, via a transaction to layer one, to a batch of transactions to be executed. This batch is made up of layer two transactions. A state proposer executes the transactions in a batch (in the order provided by the sequencer's commitment) and computes new state roots which are

written to layer one. Verifiers in a rollup ensure that state roots are (eventually) correct. Rollups come in two major types, which may change the responsibilities of these components: *optimistic* and *zero-knowledge*.

An *optimistic* rollup is one in which the state proposer is bonded and proposes new state roots. The state proposer is bonded in the sense that they stake some funds on layer one that are lost if they post an incorrect state root. A verifier in an optimistic rollup is an actor who submits a so-called *fraud proof* to challenge an incorrect state root proposed by the state proposer. In an optimistic rollup, state roots are considered correct unless a bonded verifier successfully challenges the state root with a fraud proof within some period of time (e.g., seven days). Such a verifier may need to propose an alternative state root than the one provided by the state proposer. Often, verifiers are part of the state proposer, and the combined entities are called *validators*. A verifier who successfully challenges the state proposer wins the state proposer's bond; those that lose the challenge give up their own stake to the state proposer. Arbitrum is one example of an optimistic rollup [21].

A *zero-knowledge* (ZK) rollup is one in which the state proposer generates cryptographic proofs that each new state root is correct. In a ZK rollup, the state proposer performs state transitions within a zero-knowledge proof framework (e.g., [22]), which generates a *validity proof*: an artifact that proves that a particular function was executed with particular inputs, which resulted in the new state. These validity proofs can be verified using layer one smart contracts (as a result, the verifier actor is implicit in such rollups). A state proposer for a ZK rollup may have one or more provers as a sub-component, which generates the validity proofs given a batch of transactions and a previous state root. Note that the "zero-knowledge" aspect of these proofs are sometimes helpful for privacy, but mostly these systems are used because the proofs are also *succinct*. This property enables the proofs to be verified in a fraction of the time required to run the computation in the first place, enabling verification directly on a layer one blockchain. zkSync is one example of a ZK rollup [23].

A sequencer orders transactions for the layer two. The source of these transactions may be a user of the rollup or the layer one smart contracts of the rollup. As a result, sequencers are responsible for *cross-chain* communication, and may be considered to be a blockchain *bridge*. A bridge is a system or protocol for taking assets or blockchain state from one blockchain to another. As cryptographic assets cannot be literally moved from one blockchain to another, the bridge creates representations of assets on a *source* blockchain on a *destination* blockchain. To avoid arbitrary minting of assets, bridges have a smart contract on the source blockchain called the *custodian*, which locks up the asset to be minted on the destination chain. Through an off-chain *communicator* component, when assets are placed in the custody of the bridge, the corresponding *debt issuer* on the destination blockchain mints a representation of the asset in custody. The process is reversible. Advanced bridges relay instructions to execute functions on either blockchain.

## 3 ATTACKS & RISKS

In this section, we consider attacks related to rollups. Each attack is presented in its own section.

---

[1]Other work like [15, 20] use different terms for these components, but each rollup has some component that performs these actions.

## 3.1 Censorship Attack

Ideally rollups would be as at least as censorship resistant as their underlying layer one. A rollup may be more censorship resistant because underlying layer one nodes might be easily convinced to censor transactions from a particular address, but not if that address updates an entire layer two or if the rollup can change it's underlying address on layer one when blacklisted. However, censorship can occur from several different components in a rollup and some transactions on the layer two could be withheld never be processed. In some cases, transactions can be ignored by sequencers or be excluded from the state update by the state proposer.

Misbehaving sequencers can choose to censor transactions submitted by users; this is called a *censorship attack*. If there is only one sequencer (or a small number of them), then this censorship can be very effective, and vulnerable decentralized sequencer protocol may also allow for censorship. This is partially why many rollups implement methods to forcefully include a transaction via the layer one smart contracts. However, these implementations have varying degrees of effectiveness and may not completely mitigate censorship by the sequencer.

Furthermore, some optimistic rollups have not yet fully implemented fraud proofs. Such rollups allow submitting undisputed fraudulent transaction batches, and their resulting state roots, to layer one. These state proposers may exclude transactions from the state they propose back to layer one in order to censor users. In such a system, as there is no way to successfully dispute the fraudulent state, it will be finalized on layer one. However even if fraud proofs are implemented, there is a risk of fraudulent states going unchallenged. This can be achieved by colluding state proposers or a lack of incentive to challenge invalid states. To reduce the trust required in state proposers and prevent censorship, some rollups implement a forced withdrawal functionality, where one can always force the rollup to return funds it owns.

In ZK rollups, state proposers can only censor all transactions committed by the sequencer, or none. This is because a block is required as input to the verifier when the block's validity proof is executed, and as a result, a prover cannot simply ignore a part of the input (assuming the verifier and prover are correctly implemented). Moreover, as a block includes a reference to the prior block in the chain, entire blocks also cannot be skipped. Therefore, censorship implies that no state update is posted ever again in such a ZK rollup. In a ZK rollup, as long as one honest state proposer is (eventually) online, state will progress and censorship is not possible, provided that the sequencer is not censoring transactions.

This attack can be mitigated entirely or in part by forced inclusion and forced withdrawal functionality, which are implementations of escape hatches, as the attack itself resembles an offline operator. The Hermez project previously suggested a mechanism to mitigate this attack by rotating the sequencer role according to some staked funds [16].

## 3.2 Delay Attack

A *delay attack* involves the delay of layer two state confirmation on layer one [24]. This includes intentional delays and delays incurred due to a lack of incentive. Multiple misbehaving components of rollups could cause a delay in state confirmation. The simplest version of this attack would be for a misbehaving centralized validator to simply not propose a rollup block to layer one. Sequencers could also perform delay attacks by censoring or excluding transactions from the layer two blocks for a limited time. This attack can also be mitigated entirely or in part by escape hatch functionality as it resembles an offline operator.

The risk, feasibility, and impact of delay attacks is higher in optimistic rollups compared to ZK rollups. This is due to the expanded capabilities of state proposers in optimistic rollups. In ZK rollups, state proposers can only impede state progression by refraining from submitting states for a designated delay period. They are incapable of delaying individual transactions, as the absence of a specific transaction would cause their validity proof to fail. State proposers on optimistic rollups can instigate transaction delays in two additional ways: they can omit transactions in their state update until this invalid state is contested, or they can delay resolving disputes to instigate fraud proof delays. These two delays form another attack vector which state proposers in optimistic rollups can perform, in addition to the attack vector possible on ZK rollups.

Optimistic rollups with a permissionless validator role may introduce additional risk for delays. Malicious validators may sacrifice their stake deliberately by losing challenges in order to delay the proposal and confirmation of correct layer two batches. The possibility of this attack is what currently prevents Arbitrum from making the validator role permissionless [25].

In ZK rollups, delays can occur when offline or malicious state proposers do not submit states anymore. Delayed transactions, as opposed to censored ones, will eventually change the state; state proposers can only delay all transactions or none of them since they can only use all of the transactions in a block or none of them. As long as there is one honest and online state proposer, the state is guaranteed to progress eventually.

In both kinds of rollups, the impact and risk of delay attacks depends again on the existence of escape hatch functionality so that a user can circumvent state proposers and incentives to make sure that deliberately doing the wrong thing is expensive.

## 3.3 Denial of Service Attack

A *Denial of Service (DoS)* attack involves attackers trying to prevent any action from being taken on the protocol. DoS attacks are particularly concerning as many rollups are still highly centralized and provide a single point of failure. If a single centralized actor in a rollup (i.e., sequencer, proposer, validator, or prover) no longer functions, the rollup itself could cease to function entirely. DoS concerns could be caused in many ways on rollups and we explore some example causes below.

- **Malicious Pausing.** Many rollups have some type of pause feature on their primary contracts to pause all functionality (i.e., revert on all meaningful transactions). A malicious activation of the paused state would prevent the rollup from being used. Such an approach may be feasible as access controls on smart contracts are not always easy to get right due to a lack of specifications [26].

- **Consensus Concerns.** For rollups with decentralized sequencers (that is, a possibly permissionless protocol that anyone can follow to become a sequencer), there are concerns with consensus protocol limits. Some possible consensus protocols have scalability limits, after which the network will either be unusable or face unexpected consequences (see e.g., [27] for a case study, and [28] for a survey).
- **(ZK rollups only) Expensive Proof Flooding.** For ZK rollups, some operations may be more slow to prove than others; for example, executing a hash function like SHA-256 which is not easily executed inside a proof system operating over a finite field [29]. ZK rollups which aim to be EVM equivalent are required to implement this functionality, and if it is slow to prove by the system but cheap to call on the rollup, there may be opportunities for malicious users. In particular, flooding the network with such operations may degrade the quality of service by delaying block proofs. Note that some ZK rollups may avoid this problem by replacing the the hash function or other expensive operations with more efficient versions of them.
- **(Optimistic rollups only) Fraud Proof Denial.** For optimistic rollups, preventing a validator from submitting a fraud proof would result in the incorrect outcome of the challenge game. If this can be achieved by flooding the underlying layer one with transactions with high gas fees such that the fraud proof is never included in a block until the timeout expires, the rollup will suffer from an incorrect-but-final outcome on the proof. In turn, the system state may not be one that could be achieved by honest actors; this is why optimistic rollups have a seven day challenge period [30].

## 3.4 Forged Transaction Attack

A *forged transaction attack* involves a state proposer including a fake transaction into the rollup batch. A fake transaction is a transaction that lacks a valid signature or represents a layer one event, such as a deposit, that did not take place. In an ideal rollup, this is not possible because honest users (or the verification smart contracts) can check that a layer two block containing a deposit has a layer one event recorded on a smart contract corresponding to that deposit. However, optimistic rollups which do not have a fraud system active or implement a poor incentive system to challenge illegitimate state updates could be susceptible to this kind of attack. Currently, such rollups place a high degree of trust in the proposer, which is controlled by a known entity. If such a proposer were compromised or malicious, the effects would be devastating and may lead to the loss of user funds, as an attacker could forge deposit transactions to themselves with arbitrary value. In turn, they can withdraw these funds and empty the bridge contracts of the funds.

## 3.5 Reorganization attack

Sequencers and validators must be able to account for a *reorganization* (or *reorg*) on layer one. A reorganization is when one chain is considered canonical for a period of time, but is later replaced by a different one. Problems may occur if the sequencer submits an incorrect batch or if an assumption is made about a block. A layer one reorg may change the state of the layer two network, as the

state roots are posted on the underlying layer one. Ideally, users will not have to resubmit transactions to the sequencer when the reorg on layer one occurs.

A reorg may also result in the partial rollback of an optimistic rollup's challenge completion as well. In this case, the validator must be aware of the reorg to ensure that the challenge is completed and an insecure layer two state is not finalized on layer one; they may need to resubmit their relevant transactions.

In either case, the reorg may be malicious, and result in a *reorganization attack* if it is triggered by a malicious actor. For example, if an actor can control the consensus mechanism of the underlying layer one, or if a malicious actor can quickly respond to the short term reorgs common on most layer ones because of a poor rollup design. This scenario may be expensive on Ethereum but more achievable on other layer one blockchains. However, short-term reorganizations should be expected and may be malicious (see e.g., [31]); rollups should be able to cope with such reorganizations.

## 3.6 Soft Finality Attack

Sequencers provide *soft finality* in rollups by returning receipts to users which indicate the transaction order. This is in contrast to *hard finality* when a transaction is included on the layer two (and layer one), and a layer two state root containing that update cannot be removed from the record on layer one, either by a challenge or a (reasonably feasible) layer one reorganization. Soft finality allows users who trust the chain to make decisions related to a transaction's inclusion without waiting for hard finality.

Soft finality can be invalidated by centralized sequencers in a rollup; this is a *soft finality attack*. For example, the sequencer may order the transaction batch differently than what was indicated in the receipt or it may also exclude the transaction despite returning a receipt. Users may suffer losses or inconveniences due to assumptions made about the ordering of the transaction based on a fake receipt in either of the previous scenarios In some sense, this is their own fault — the transaction was not guaranteed to be included after all — but soft finality provides an enhanced user experience for Rollups. Moreover, violating it may be an attack in the sense that the sequencer can explicitly profit from the omission or cause particular parties to be negatively impacted.

## 3.7 Sybil Attack

Most rollups do not currently have permissionless sequencers and validators (that is, these roles are fulfilled by whitelisted actors with special permissions to fill them). To make these components permissionless, some form of consensus algorithm must be introduced to ensure that a set of actors with the same role agrees on the outcome. Attackers may perform a *Sybil attack* to control sufficient nodes: an actor may masquerade as several entities in order exploit the consensus mechanism [32].

It is essential that the barrier is low for participating as a sequencer or validator in order to sufficiently decentralize the rollup. Otherwise, the few that are capable of participating may collude or create multiple identities to execute a Sybil attack. However, a low barrier for participation may encourage Sybil attacks, so it is important that there are protocol-level measures to safeguard against them, too.

## 3.8 Client Vulnerability Risk

The robustness of the Ethereum blockchain is in part due to the diversity of clients implementing its design. Even if one client software has a bug, the nodes running another client may compensate for this. Rollup nodes run their own client, which is distinct from Ethereum clients (though it may be based on one of them). Bugs in the implementation of this client can lead to a loss of funds in the worst case (see e.g., [33]) and disruption of the rollup functionality in other cases.

Client diversity should extend to L2s for both sequencers and validators. This is because a bug in the implementation of the fault or validity proof system affects safety of the rollup [34]. For example, if a bug-ridden state proposer attempts to post a state root in an optimistic rollup that is actually incorrect due to a bug, a validator with a client that does not contain that bug will correctly challenge the state root (and ideally win). Naturally, each client should also be thoroughly tested and reviewed to minimize the presence of bugs.

## 3.9 Accountability Attack

In [35], *accountability* of a rollup is summarized as follows

> Accountability requires that whenever the rollup clients obtain commitments to conflicting rollup states, the staked nodes of the [layer one] responsible for this safety breach must be identified and slashed. However, since the clients do not download the [layer one] data that is not relevant to the rollup, they can be tricked into accepting rollup transactions within unavailable [layer one] blocks.

An *accountability attack* is one that occurs after another safety violating attack occurs; essentially, one in which the rollup operators used unavailable layer one blocks to derive their state. In [35] it is also shown that rollups using Ethereum as the layer one may be susceptible to such an attack. This attack results in a situation where "no adversarial [layer one] node can be provably accused" and does not necessarily cause direct loss of funds itself on the layer two, though it may result in a layer two reorganization.

## 4 ROLLUP SECURITY IMPROVEMENTS

We stress the following areas for future development and work on rollups in order to improve their security.

*Always-On, Scalable Escape Hatches.* Rollups should have efficient, always-on escape hatches [20]. They should not be pausible, they should be scalable, and they should be usable.

Escape hatches should be feasible and accessible when necessary. Some rollups such as dYdX have no mechanism to allow operators to disable the escape hatch. However, general purpose chains have mechanisms which can prevent users from accessing the escape hatch. For example, it is possible for the designated multi-sig to pause Arbitrum's DelayedInbox contract (a part of the escape hatch functionality) or restrict its usage to users on a whitelist. This would remove users' ability to access the escape hatch and greatly increase censorship risk.

Many existing escape hatches involve users submitting transactions directly on layer one, except those which allow new state proposers. These transactions may only be able to transfer one digital asset off of the chain at a time, though a user may have many.

User should have to execute as few of them as possible to exit the rollup in order to provide an good user experience and to avoid paying high layer one transaction fees. Moreover, an asset may be in a layer two Decentralized Finance (DeFi) protocol [36] which must be forcefully withdrawn first, adding an additional layer one transaction to the process. If each of these operations requires a different layer one transaction, the cost of exiting may be prohibitive. One possible improvement is to allow a forced transaction to take multiple actions on the layer two prior to the withdrawal action, reducing the number of layer one transactions necessary.

Finally, we note that there are currently no dedicated front-end interfaces for users to access escape hatches. Users must interact with smart contracts directly in order to use the escape hatches. Even somewhat technical users may be confused as to what parameters to use in the escape hatch functions. For example, the sendUnsignedTransaction() function in Arbitrum's DelayedInbox requires the parameters: gasLimit, maxFeePerGas, nonce, to, value, and data. Many users would not know how to use this function in order to perform even a simple withdrawal a token from layer one to layer two. As a result of the expertise required to use these escape hatches, they remain inaccessible.

*Parallel or "All-vs-All" Challenges in Optimistic Rollups.* In optimistic rollups, the transition from centralized to decentralized state proposers is ongoing but constrained by the vulnerability to delay attacks due to extended dispute periods (see Section 3.3). As rollups systems make strides towards decentralization, they need to implement robust measures to prevent these attacks.

The introduction of "all-vs-all" challenges presents an innovative solution to this problem. This mechanism, as currently under development in Arbitrum [25], restricts the maximum delay a potential attacker can cause to a disputed block, irrespective of the stakes they are willing to sacrifice. In the context of fraud proofs, "all-vs-all" challenges empower even a single honest staker to efficiently neutralize a multitude of attackers who may deliberately loose while defending honest states or post malicious branching assertions. This provides a significant fortification against delay attacks (Section 3.2), thereby enabling a more secure path towards the decentralization of state proposers on optimistic rollups.

*Clear Documentation of Access Control Scope.* The current landscape of rollup solutions underscores a significant need for more comprehensive documentation of access control scoping, which includes areas such as operator assignment and system upgrades. The control of operator roles is a critical aspect that is usually not well conveyed to the users. This includes questions around whether escape hatches can be deactivated, contracts can be upgraded, and how this role changes hands, who has the authority to perform each action, and under what circumstances an action can be taken. For example, if rollup operators can control the escape hatches, e.g., by deactivating them, this needs to be communicated to the users as they should make the deliberate decision to trust the rollup operator, and users should determine if the operator is trustworthy and the risk is sufficiently low.

## 5 CONCLUSION

We have reviewed attacks on rollups, considering their impact on both optimistic and ZK rollups. After we defined each attack, we

illustrated potential impacts of it. Future work should consider comprehensive solutions to mitigate these attacks and consider rollup variations, like so-called *enshrined* rollups which are more tightly coupled to their underlying layer one or *sovereign* rollups which post data elsewhere. As rollups continue to be an increasingly popular form of middleware on general purpose blockchains, these attacks need to be well understood in order to prevent their exeuction in real-world settings.

## ACKNOWLEDGMENTS

This work was funded in part by the Ethereum Foundation through Ecosystem Support Program grant FY23-0898. The authors would like to thank the anonymous reviewers and the security researchers at Quantstamp for their helpful comments.

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
