# OpenReview forum: "Attacks on Rollups"
_ACM.org/Middleware/Workshop/DICG — DICG 2023_

### Official Review · Reviewer_hpcW · 2023-10-27
**Review of Attacks on Rollups**

**Rating:** 8
**Confidence:** 5

**Review:**

This paper examines potential attack vectors on rollup solutions. The authors first provide a general introduction to rollups and then delve into attack vectors, discussing their feasibility, impact, and possible mitigations. Finally, the authors propose some security design principles for rollups.

The paper effectively addresses an emerging concept in blockchain technology. It provides comprehensive coverage of potential attacks and, given the space limitations, discusses them concisely yet thoroughly. Given the importance of scalability in permissionless distributed ledgers, the paper is a good fit for the workshop and has the potential to become a decent reference in the field. My only suggestion is to consider mentioning "enshrined rollups," which could potentially mitigate some of the attacks discussed in the paper. I recommend a minor revision of the paper for the camera-ready version, addressing the considerations below.

### Additional comments:

- In the abstract, the phrase "the state of the rollup network" may be misleading. It might be more accurate to say "the state of the log that the network maintains."
- In the abstract, the phrase "meaning that the rollup state cannot be changed arbitrarily" can be misleading, as the loss of transactions can be recorded on L1 during the state commitment. A phrase like "as the submitted rollup state is verified on the underlying blockchain, arbitrary changes can be detected" would be more accurate.
- On page 2, in the sentence just before the last paragraph, it's mentioned that "zkSync is an example of ZK rollup." It would be beneficial to also mention other well-known zk rollup solutions for reference, such as StarkNet, Scroll, Polygon zkEVM, Taiko, etc.
- In Section 3.1, the statement "Ideally rollups would be as censorship resistant as their underlying layer one" could be revised to acknowledge that a rollup might actually be more censorship resistant than its underlying layer one. In theory, a transaction that is censored by L1 nodes could find a place on a completely centralized rollup, as it would be less feasible for L1 nodes to censor the entire rollup state to block a single transaction.
- In Section 3.1, the argument that "In a ZK rollup, as long as one honest state proposer is (eventually) online, state will progress and censorship is not possible" should be clarified to indicate that this argument assumes the sequencer does not engage in censorship. The current sentence may give the impression that if there is one honest state proposer, the ZK rollup is censorship-resistant, which is not entirely accurate since the sequencer can still engage in censorship.
- In Section 3.3, regarding "Expensive Proof Flooding," it would be helpful to mention that some zk rollup solutions replace these costly calculations with more efficient operations, as seen in zkSync.
- In Section 3.6, if I'm not mistaken, the phrase "when a transaction is included on the layer two" might need to be revised to "when a transaction is included on the layer one."
- The second paragraph of Section 3.7 may need revision. Lowering the barrier to entry may, in turn, encourage attackers to create more Sybil entities. The key concern here is resistance to Sybil attacks, regardless of the entry barrier.

### Minor comments:

- In Section 3.2, an incomplete sentence: "since they can only all of the inputs to a block or none of them."
- Typos:
    - Two occurrences of "an layer one" --> "a layer one."
    - "finalityin" --> "finality in."
- Grammar:
    - "this is not be possible"

---

### Official Review · Reviewer_odxw · 2023-10-30
**Review of "Attacks on Rollups"**

**Rating:** 7
**Confidence:** 5

**Review:**

This paper provides a succinct and useful review of the attacks on the rollups, which are an emerging type of “middleware” that seek to improve the speed of transactions on the main network. Rollups are typically executed away from the main network (i.e., off-chain), which is partly the cause of the various possible types of attacks.

The paper is well written, and in describing each attack type the paper goes directly to heart of the cause/reason why it can occur (e.g., dishonest sequencer). So it provides a very nice short summary of each type of attack.

The paper has a good coverage of references, and in many cases identifies the systems that implement some of the rollup features and thus maybe susceptible to certain attacks.

The paper mentions directions for research into improving these designs (e.g., escape hatches). However, since the goal of the paper is to provide a literature review, we do not expect solutions to be discussed.

All in all, it is an easy paper to read and provides a useful guide for readers researching rollups, layer-2 and bridges.